

# Predictive value of CONUT score combined with serum CA199 levels in postoperative survival of patients with pancreatic ductal adenocarcinoma: a retrospective study

Ankang Wang[1,2,*], Bo Sun[1,*], Min Wang[3], Hao Shi[1], Zhiwei Huang[1], Tao He[1], Qiu Li[1], Jiaqi Deng[4], Wenguang Fu[1] and Yu Jiang[1]

[1] Department of Hepatobiliary Surgery, The Affiliated Hospital of Southwest Medical University, Sichuan, China
[2] Department of General Surgery, Nanchong Central Hospital, The Second Clinical College of North Sichuan Medical College, Nanchong, Sichuan, China
[3] Department of Nutrition, The Affiliated Hospital of Southwest Medical University, Luzhou, Sichuan, China
[4] Department of Ultrasound, The Affiliated Hospital of Southwest Medical University, Sichuan, China
* These authors contributed equally to this work.

Corresponding authors
Wenguang Fu, fuwg@swmu.edu.cn
Yu Jiang, 415943914@qq.com

## ABSTRACT

**Background:** The preoperative controlling nutritional status (CONUT) score and serum carbohydrate antigen 199 (CA199) levels are individually correlated with the prognosis of pancreatic ductal adenocarcinoma (PDAC). The objective of this study aimed to investigate the efficacy of CONUT score and CA199 (CONUT-CA199) combination in predicting the prognosis of PDAC patients undergoing radical surgery.

**Methods:** We retrospectively analyzed the preoperative CONUT scores and serum CA199 levels of 294 patients with PDAC who underwent radical resection at the Affiliated Hospital of Southwest Medical University between March 2012 and July 2019. Patients were divided into four groups on the basis of their preoperative CONUT scores and serum CA199 levels: CONUT$^{low}$/CA199$^{low}$ (1), CONUT$^{low}$/CA199$^{high}$ (2), CONUT$^{high}$/CA199$^{low}$ (3) and CONUT$^{high}$/CA199$^{high}$ (4). The prognostic effects were compared among the groups.

**Results:** CONUT$^{high}$ was more frequent in patients with positive peripancreatic infiltration and Clavien–Dindo classification of ≥IIIa ($P < 0.001$). Kaplan–Meier analysis revealed obvious difference in overall survival (OS) and recurrence-free survival (RFS) among patients with PDAC having CONUT-CA199 scores of 1, 2, 3 and 4 ($P < 0.001$). Peripancreatic infiltration, lymph node metastasis, pTNM stage, CONUT score, serum CA199 levels and CONUT-CA199 classification were found to be the independent prognostic factors for OS and RFS in multivariate analyses. In time-dependent receiver operating characteristic (ROC) analyses, the area of the CONUT-CA199 score under the ROC curve (AUC) was higher than that of the preoperative CONUT score or serum CA199 levels for the prediction of OS and RFS.

**Conclusion:** CONUT-CA199 classification may be more effective in predicting the postoperative prognosis of PDAC patients.

antigen 199, Prognosis

## INTRODUCTION

Pancreatic cancer (PC) is a malignant disease with strong invasiveness in humans and it is expected to be the second leading cause of cancer related death in the future (*Rahib et al., 2014*). Pancreatic ductal adenocarcinoma (PDAC), which originates from the ductal epithelium, is the most common histological type of PC, accounting for approximately 95% of pancreatic exocrine tumors (*Kamisawa et al., 2016*). Currently, the only available treatment modality for PDAC is surgical resection (*Gong et al., 2013*). However, the disease is often diagnosed at a later stage owing to its initially unpredictable biological behavior. At that time, liver, lymph nodes, peripheral vessels and nerves are often affected, with the tumor showing rapid growth, resulting in poor prognosis and the 5 year survival rate has been stagnant at 6% for decades (*El-Khayat et al., 2018*; *Siegel, Miller & Jemal, 2019*). Numerous studies have shown that tumor size, lymph node metastasis, vascular invasion and serum tumor markers (TMs) are vital prognostic factors for PC (*Karamitopoulou et al., 2013*; *Staal et al., 2019*; *Winter, Yeo & Brody, 2013*). Moreover, early detection of postoperative recurrence can help improve the survival rate of patients with PDAC (*Wu et al., 2019*); therefore, it is important to determine the factors affecting the prognosis of these patients after pancreatectomy. It is not difficult to obtain the serum TMs level of patients from clinical data, which is of potential value for diagnosis, monitoring of postoperative recurrence and predicting survival rate (*Fujioka et al., 2007*). The serum marker CA199 has shown diagnostic potential in patients with latent and early PDAC (*Haab et al., 2015*; *O'Brien et al., 2015*) and can predict disease progression (*Duffy et al., 2010*; *Satake et al., 1985*).

Relevant reports have shown that the prognosis of tumor is closely related to the inflammatory status, immune function and nutritional status of patients (*Mantzorou et al., 2017*; *Ni et al., 2019*; *Van Dijk & Pot, 2016*; *Xiao et al., 2019*). Numerous studies have found that malnutrition significantly increases postoperative complications and has a negative impact on the quality of life, hospital stay, anti-cancer treatment effect and overall survival in cancer patients (*Borre et al., 2018*; *Fujiya et al., 2018*; *Lin et al., 2019*). Furthermore, the latest research shows that there is a close relationship between nutritional status and prognosis of patients with cancer, including PDAC (*Abe et al., 2018*; *Balzano et al., 2017*; *Gilliland et al., 2017*). The controlling nutritional status (CONUT) score, a system for scoring immune nutritional status that emerged in 2005, has garnered the attention of researchers (*De Ulibarri et al., 2005*), it includes the measurement of serum albumin and total cholesterol levels as well as peripheral blood lymphocytes. This scoring system has been considered as a predictor of prognosis for postoperative liver cancer, gastric cancer, colorectal cancer and PDAC (*Iseki et al., 2015*; *Kato et al., 2018*; *Shoji et al., 2017*; *Takagi et al., 2017*). The patient's serum TMs are mostly determined by the tumor itself, whereas the CONUT score reflects the overall immune and nutritional status of patients. Both indicators demonstrate their role in assessing the prognosis of patients with

PDAC. However, the value of their joint application is still unclear, this study aims to use these two indicators in combination to evaluate the prognosis of patients with PDAC.

## MATERIALS AND METHODS

### Study population

All patients with PC who received radical resection in the affiliated hospital of the Southwest Medical University between March 2012 and July 2019 were retrospectively analyzed; a total of 294 cases met the inclusion criteria of this study. Inclusion criteria were as follows: patients (1) with histopathological confirmation of PDAC; (2) who had undergone radical resection; (3) who did not receive any neoadjuvant chemotherapy and/or radiotherapy before surgery; (4) with no history of other malignant tumors; (5) with complete clinical and follow-up data; and (6) in whom no metastatic lesions were found in the whole body before surgery. Exclusion criteria were as follows: patients (1) with acute or chronic infectious diseases preoperatively; (2) with preoperative complications of blood system diseases, kidney diseases, or cardiovascular and cerebrovascular diseases; (3) with any other known autoimmune disease; (4) with history of steroid medication use within 15 days before operation; (5) who received preoperative immune enhancement therapy or had a recent history of blood transfusion; and (6) who died within 30 days after operation.

### Investigational variables

All preoperative clinicopathological data were obtained from the electronic medical record system; the data included age, gender, height, weight, serum CA199 levels and carcinoembryonic antigen (CEA), tumor location, tumor size, histopathological type. Invasion and metastasis of peripancreatic, lymph nodes, lymphatic vessels and blood vessels and pTNM staging were performed. Prognostic nutrition index (PNI) and CONUT score were calculated by blood routine results. Complications were presented by Clavien–Dindo classification and incidence of pancreatic fistula. Blood samples were collected 1 week before surgery and assessed for serum albumin and total cholesterol levels as well as total peripheral lymphocyte count. CONUT scores and PNI (*De Ulibarri et al., 2005*; *Pinato, North & Sharma, 2012*) were calculated according to previously described methods, as shown in Table 1. Postoperative complications were presented by the Clavien–Dindo classification system (*Clavien et al., 2009*) and the 2016 version of the postoperative pancreatic fistula grading system released by the International Study Group on Pancreatic Surgery (*Bassi et al., 2017*). The largest diameter of the tumor in the pathological sampling was considered as tumor size and tumor staging was performed according to the TNM staging criteria of AJCC version 8 (*Van Roessel et al., 2018*). We communicated with the patients before surgery and their consent was orally obtained for our study. Our research was supported by the Ethics Committee of the Affiliated Hospital of Southwest Medical University (No. KY2019053).

### Follow-up

We follow up all patients in a standardized way. Follow-up examination included abdominal ultrasound, chest X-ray, routine blood work, blood biochemistry

**Table 1 Scoring system for the controlling nutritional status (CONUT).**

| Degree of undernutrition | CONUT score | Serum albumin (g/dl) | Total lymphocyte (/mm$^3$) | Total cholesterol (mg/dl) |
|---|---|---|---|---|
| Normal | 0–1 | ≥3.50 (0) | ≥1,600 (0) | ≥180 (0) |
| Mild | 2–4 | 3.00–3.49 (2) | 1,200–1,599 (1) | 140–179 (1) |
| Moderate | 5–8 | 2.50–2.99 (4) | 800–1,199 (2) | 100–139 (2) |
| Severe | 9–12 | <2.50 (6) | <800 (3) | <100 (3) |

Note:
   CONUT score = Serum albumin score + total lymphocyte score + total cholesterol score.

(liver function, renal function) and TMs assessment. Contrast-enhanced computed tomography, magnetic resonance imaging, positron emission tomography and other modalities were used depending on the situation if a suspicious lesion was detected and the nature of the lesion could not be defined. In accordance with the Chinese comprehensive guidelines for the diagnosis and treatment of PC (*Pancreatic Cancer Committee of Chinese Anti-Cancer A, 2018*), the patients were reexamined every 3 months during the first year after surgery, followed by every 3–6 months in next 2–3 years. After 3 years, the follow-up period changed to 6 months. Survival data were obtained through patient outpatient visits and telephone follow-up. We counted the interval between the completion of surgery and death or the last follow-up and the interval between the completion of surgery and tumor recurrence or the last follow-up, respectively expressed as the overall survival (OS) and Recurrence-free survival (RFS). Tumor recurrence included local recurrence and distant metastasis (liver and peritoneum, lungs, bone, etc.). The follow-up deadline was August 2019.

### Definition of preoperative CONUT-CA199 score

The optimal cutoff value of preoperative CONUT score was 3, which was used as the criterion to divide 294 patients into low group (<3; $n = 194$) and high group (≥3; $n = 100$). Patients were divided into the following two groups according to the optimal cutoff value of serum CA199 levels (36.6 ng/mL): CA199$^{low}$ (<36.6; $n = 148$), CA199$^{high}$ (≥36.6; $n = 146$). Based on the cutoff values of preoperative CONUT and CA199, we defined the CONUT-CA199 score. Patients with CONUT$^{low}$/CA199$^{low}$ ($n = 95$) were assigned a score of 1; those with CONUT$^{low}$/CA199$^{high}$ ($n = 99$) were assigned a score of 2; those with CONUT$^{high}$/CA199$^{low}$ ($n = 53$) were assigned a score of 3; and those with CONUT$^{high}$/CA199$^{high}$ ($n = 47$) were assigned a score of 4.

### Statistical analyses

The classified data were summarized using a number (%) and the difference between each group of variables is detected by chi-square test. A post hoc power analysis was completed. The power of the Peripancreatic infiltration and the Clavien–Dindo classification group was 0.64 and 0.98, respectively. The optimal cutoff values of CONUT score, CA199, CEA, age, size, PNI and the area under the curve (AUC) were obtained by receiver operating characteristic (ROC) curve analysis. Survival curves were presented using the Kaplan–Meier method and the differences were compared by log-rank test.

Firstly, univariate analysis was carried out for various clinical and pathological variables and covariates with $P$ value <0.05 were included in multivariate analysis. Cox proportional hazard model and stepwise analysis were used to obtain independent influencing factors of OS and RFS. IBM SPSS Statistics package v.24.0 (Chicago, IL, USA) was used for statistical analysis, $P < 0.05$ was considered statistically significant.

## RESULTS

A total of 294 patients who met the criteria were enrolled (163 men (55.4%) and 131 women (44.6%); age range, 29–78 years; mean age, 55.5 ± 10.8 years).

Among the enrolled patients, 214 (72.8%) had tumors in the pancreatic head, 63 (21.4%) had tumors in the pancreatic body and tail and 17 (5.8%) had tumors that were diffuse in the pancreas. Among all patients, 131 (44.5%) had poorly differentiated, 96 (32.7%) had moderately differentiated and 67 (22.8%) had highly differentiated tumors. There were 70 (23.8%), 125 (42.5%) and 99 (33.7%) patients with stage I, II and III tumors, respectively. The general situation of the two groups of patients is shown in Table 2. CONUT$^{high}$ was more frequent in patients with positive peripancreatic infiltration and Clavien–Dindo classification ≥IIIa ($P < 0.001$).

The 5 year OS of the CONUT$^{low}$ group (11.0%) was significantly higher than that of the CONUT$^{high}$ group (2.9%) ($P < 0.0001$) (Fig. 1A). The 5 year OS rate of the CA199$^{high}$ group (4.7%) was lower than that of the CA199$^{low}$ group (13.3%) ($P < 0.013$) (Fig. 1B).

Patients were divided into four groups to determine the impact of combining the CONUT scores and serum CA199 levels (CONUT-CA199) on prognosis. The 5 year OS rates of patients with CONUT$^{low}$/CA199$^{low}$, CONUT$^{low}$/CA199$^{high}$, CONUT$^{high}$/CA199$^{low}$ and CONUT$^{high}$/CA199$^{high}$ were 15.3%, 9.1%, 6.1% and 0%, respectively ($P < 0.0001$) (Fig. 2A). In addition, the similar 5 year RFS rates were 9.2%, 7.9%, 4.7% and 0%, respectively ($P < 0.0001$) (Fig. 2B). ROC analysis was used to further evaluate the effect of three independent factors on prognosis in our research. The results showed that the preoperative CONUT-CA199 scores were more predictive of OS and RFS in patients with PDAC than preoperative CONUT scores or preoperative serum CA199 levels alone (OS: AUC = 0.685 (95% CI [0.625–0.746]); $P < 0.001$; RFS: AUC = 0.692 (95% CI [0.632–0.751]); $P < 0.001$; Figs. 3A and 3B).

Univariate analyses showed that age (<52 vs. ≥52 years; $P < 0.05$), serum CA199 levels (<36.6 vs. ≥36.6 ng/mL; $P < 0.001$), tumor size (<3.1 vs. ≥3.1 cm; $P < 0.05$), histopathological type (poorly differentiated vs. moderate-highly differentiated; $P < 0.001$), peripancreatic infiltration (positive vs. negative; $P < 0.001$), lymph node metastasis (positive vs. negative; $P < 0.001$), superior mesenteric artery invasion (positive vs. negative; $P < 0.001$), portal vein system invasion (positive vs. negative; $P < 0.05$), nerve plexus invasion (positive vs. negative; $P < 0.05$), pTNM stage (I/II vs. III; $P < 0.001$), PNI (<46.1 vs. ≥46.1; $P < 0.001$), the CONUT score (low vs. high; $P < 0.001$), the CONUT-CA199 score (1 vs. 2 vs. 3 vs. 4; $P < 0.001$) were related to OS and RFS (Table 3).

Since the CONUT-CA199 score includes the CONUT score and serum CA199 levels, two multi-factor Cox proportional models were set up to avoid colinearity problems. Among them, peripancreatic infiltration ($P < 0.05$), lymph node metastasis ($P < 0.001$),

**Table 2 Relationships between CONUTscore and clinicopathological characteristics of 294 PDAC patients.**

| Variable | CONUT$^{low}$ ($n$ = 194) | CONUT$^{high}$ ($n$ = 100) | $\chi^2$ value | $P$ value |
|---|---|---|---|---|
| Gender | | | 0.401 | 0.526 |
| Male | 105 (54%) | 58 (58%) | | |
| Female | 89 (46%) | 42 (42%) | | |
| Age (years) | | | 3.205 | 0.073 |
| <52 | 87 (45%) | 34 (34%) | | |
| ≥52 | 107 (55%) | 66 (66%) | | |
| BMI (kg/m$^2$) | | | 3.455 | 0.178 |
| <18.5 | 36 (19%) | 25 (25%) | | |
| ≥18.5, <25.0 | 128 (66%) | 61 (61%) | | |
| ≥25.0 | 30 (15%) | 14 (14%) | | |
| Tumour location | | | 3.036 | 0.219 |
| Pancreatic head | 145 (75%) | 69 (69%) | | |
| Pancreatic body and tail | 41 (21%) | 22 (22%) | | |
| Dispersed | 8 (4%) | 9 (9%) | | |
| Tumor size (cm) | | | 1.925 | 0.165 |
| <3.1 | 134 (69%) | 61 (61%) | | |
| ≥3.1 | 60 (31%) | 39 (39%) | | |
| Histopathological type | | | 2.546 | 0.111 |
| Poorly differentiated | 80 (41%) | 51 (51%) | | |
| Medium-high differentiation | 114 (59%) | 49 (49%) | | |
| Peripancreatic infiltration | | | 4.447 | 0.035* |
| Positive | 144 (74%) | 85 (85%) | | |
| Negative | 50 (26%) | 15 (15%) | | |
| Lymph node metastasis | | | 0.866 | 0.352 |
| Positive | 84 (43%) | 49 (49%) | | |
| Negative | 110 (57%) | 51 (51%) | | |
| Lymphatic vessel invasion | | | 0.365 | 0.546 |
| Positive | 146 (75%) | 72 (72%) | | |
| Negative | 48 (25%) | 28 (28%) | | |
| Invasion of portal vein system | | | 2.356 | 0.125 |
| Positive | 53 (27%) | 36 (36%) | | |
| Negative | 141 (73%) | 64 (64%) | | |
| Superior mesenteric artery invasion | | | 1.522 | 0.217 |
| Positive | 58 (30%) | 37 (37%) | | |
| Negative | 136 (70%) | 63 (63%) | | |
| Nerve plexus invasion | | | 1.081 | 0.299 |
| Positive | 120 (62%) | 68 (68%) | | |
| Negative | 74 (38%) | 32 (32%) | | |
| pTNM stage | | | 3.642 | 0.056 |
| I–II | 136 (70%) | 59 (59%) | | |
| III | 58 (30%) | 41 (41%) | | |

| Variable | CONUT$^{low}$ (n = 194) | CONUT$^{high}$ (n = 100) | $\chi^2$ value | P value |
|---|---|---|---|---|
| Clavien–Dindo classification | | | 24.342 | <0.001* |
| <IIIa | 170 (88%) | 63 (63%) | | |
| ≥IIIa | 24 (12%) | 37 (37%) | | |
| Pancreatic fistula | | | 0.269 | 0.604 |
| Presence | 34 (18%) | 20 (20%) | | |
| Absence | 160 (82%) | 80 (80%) | | |

Notes:
* $P < 0.05$.
PDAC, Pancreatic ductal adenocarcinoma; BMI, body mass index; pTNM, Pathologic tumor-node-metastasis; CONUT, controlling nutritional status; The cut off value of CONUT score is 3, according to the ROC analyses.

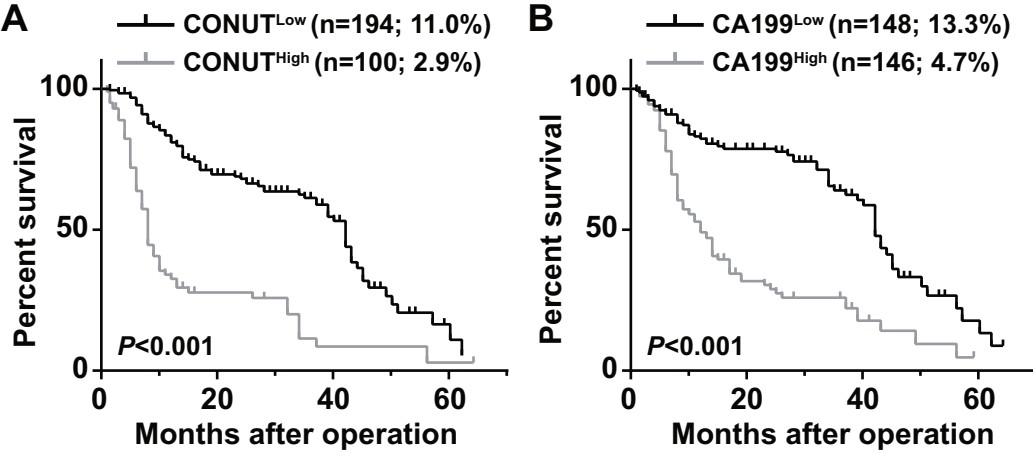

**Figure 1 Overall survival curves for pancreatic ductal adenocarcinoma patients according to CONUT score (A) and serum CA199 level (B).** CONUT, Controlling Nutritional Status; CA199, carbohydrate antigen 199. 

pTNM stage ($P < 0.05$), the CONUT score ($P < 0.001$), serum CA199 levels ($P < 0.001$) and the CONUT-CA199 score ($P < 0.001$) were independent prognostic factors for OS and RFS in multivariate analyses (Table 4).

# DISCUSSION

The CONUT score has been suggested as an indicator of immune-nutritional status of the host (*De Ulibarri et al., 2005*; *Tokunaga et al., 2017*). Increasing body of documents have suggested that patients with high preoperative CONUT scores generally have poor nutritional and pro-tumor immunity status, potentially leading to tumor invasion and metastasis. A growing number of studies have shown that patients with high preoperative CONUT score are generally poorer in nutritional status and pro-tumour immunity status and promote tumor invasion and metastasis (*Liang et al., 2017*; *Shoji et al., 2017*), which is significant for survival prognosis in postoperative patients with multiple cancers (*Harimoto et al., 2018*; *Liu et al., 2018*; *Yang et al., 2019*). Related studies have shown

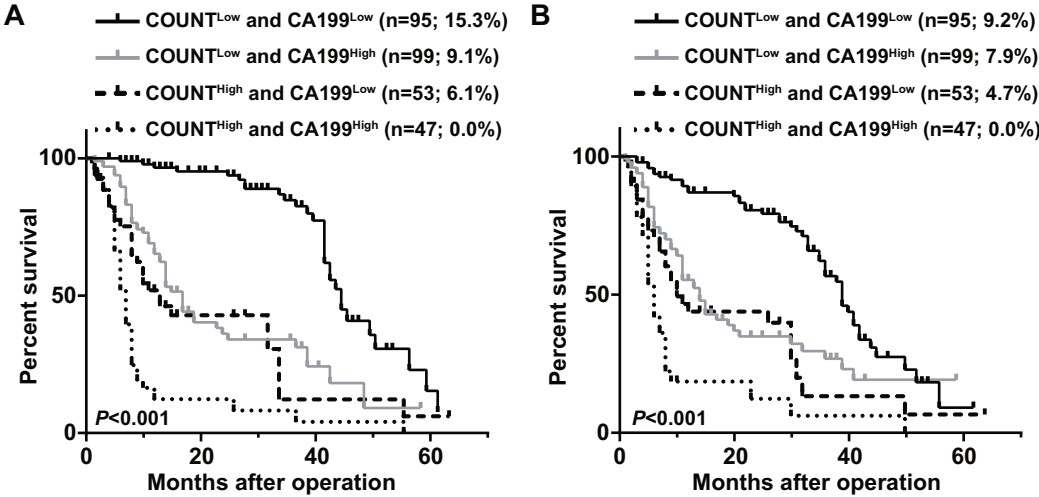

**Figure 2 Overall survival curves (A) and recurrence-free survival curves (B) for pancreatic ductal adenocarcinoma patients according to the combination of CONUT score and serum CA199 level.** CONUT, Controlling Nutritional Status; CA199; carbohydrate antigen 199.

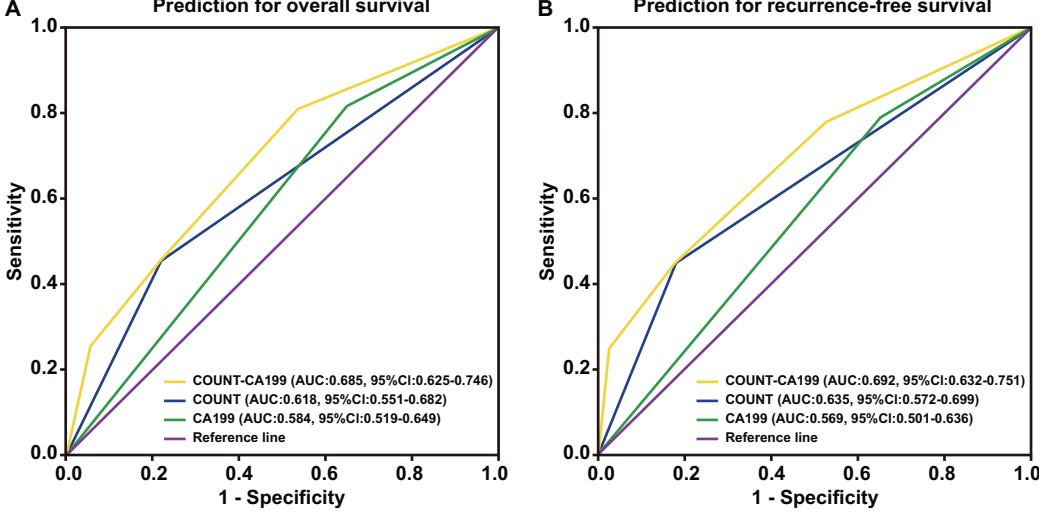

**Figure 3 Time-dependent ROCcurves of preoperative CONUT score, serum CA199 level and CONUT-CA199 scorefor the prediction of pancreatic ductal adenocarcinoma patients' outcomes.** (A) Overall survival. (B) Recurrence-free survival. CONUT, controlling nutritional status; CA199; carbohydrate antigen 199; ROC, receiver operating characteristic; AUC, area under curve; CI, confidence interval.

that the CONUT score is associated with survival prognosis of patients with unresectable PDAC and is an independent predictor of survival of patients with PDAC after pancreatectomy (*Asama et al., 2018*; *Kato et al., 2018*). Similar to previous reports, the finding of our report indicate that the preoperative CONUT score has value in predicting the postoperative prognosis of PADC patients. As the highest protein in human plasma produced by the liver, albumin can be used to assess the nutritional status of the body. Patients with low serum albumin levels are associated with poor nutritional and immune

**Table 3 Univariate analyses of factors associated with overall survival and recurrence-free survival of PDAC patients.**

| Variable | OS | | | RFS | | |
|---|---|---|---|---|---|---|
| | HR | 95% CI | *P* value | HR | 95% CI | *P* value |
| Gender (Male vs. Female) | 0.78 | [0.57–1.08] | 0.137 | 0.85 | [0.63–1.16] | 0.311 |
| Age (<52 vs. ≥52 years) | 1.51 | [1.08–2.11] | 0.015* | 1.49 | [1.09–2.03] | 0.013* |
| Preoperative CEA (<6.4 vs. ≥6.4 ng/ml) | 1.41 | [0.95–2.11] | 0.092 | 1.29 | [0.87–1.91] | 0.200 |
| Preoperative CA199 (<36.6 vs. ≥36.6 ng/ml) | 3.03 | [2.15–4.27] | <0.001* | 2.22 | [1.63–3.04] | <0.001* |
| BMI (<18.5 vs. ≥18.5, <25.0 vs. ≥25.0 kg/m$^2$) | 1.10 | [0.85–1.44] | 0.471 | 1.04 | [0.81–1.33] | 0.789 |
| Tumour size (<3.1 vs. ≥3.1 cm) | 1.49 | [1.07–2.07] | 0.018* | 1.58 | [1.16–2.16] | 0.004* |
| Tumour location (Head vs. Body and tail vs. Dispersed) | 0.89 | [0.68–1.17] | 0.402 | 0.83 | [0.63–1.08] | 0.157 |
| Histopathological type (Poor vs. Medium-high) | 0.54 | [0.38–0.75] | <0.001* | 0.55 | [0.40–0.80] | <0.001* |
| Peripancreatic infiltration (Positive vs. Negative) | 0.34 | [0.21–0.53] | <0.001* | 0.40 | [0.26–0.59] | <0.001* |
| Lymph node metastasis (Positive vs. Negative) | 0.28 | [0.20–0.41] | <0.001* | 0.31 | [0.22–0.43] | <0.001* |
| Lymphatic vessel invasion (Positive vs. Negative) | 0.92 | [0.62–1.35] | 0.657 | 0.93 | [0.65–1.33] | 0.686 |
| Superior mesenteric artery invasion (Positive vs. Negative) | 0.45 | [0.32–0.63] | <0.001* | 0.50 | [0.36–0.69] | <0.001* |
| Invasion of portal vein system (Positive vs. Negative) | 0.55 | [0.39–0.78] | 0.001* | 0.54 | [0.39–0.76] | <0.001* |
| Nerve plexus invasion (Positive vs. Negative) | 0.67 | [0.48–0.95] | 0.023* | 0.66 | [0.48–0.90] | 0.010* |
| pTNM stage (I–II vs. III) | 2.86 | [2.05–4.00] | <0.001* | 2.44 | [1.78–3.35] | <0.001* |
| Clavien–Dindo classification (<IIIa vs. ≥IIIa) | 1.54 | [1.06–2.24] | 0.123 | 1.42 | [0.99–2.04] | 0.059 |
| Pancreatic fistula (Presence vs. Absence) | 0.92 | [0.62–1.38] | 0.700 | 0.94 | [0.64–1.38] | 0.749 |
| Preoperative PNI (<46.1 vs. ≥46.1) | 0.51 | [0.37–0.71] | <0.001* | 0.58 | [0.43–0.79] | 0.001* |
| Preoperative COUNT score (Low vs. High) | 3.50 | [2.52–4.87] | <0.001* | 2.68 | [1.95–3.69] | <0.001* |
| Preoperative CONUT-CA199 score (1 vs. 2 vs. 3 vs. 4) | 2.10 | [1.81–2.44] | <0.001* | 1.78 | [1.54–2.05] | <0.001* |

**Notes:**
* $P < 0.05$.
PDAC, Pancreatic ductal adenocarcinoma; HR, hazard ratio; CI, confidence interval; OS, overall survival; RFS, recurrence-free survival; BMI, body mass index; pTNM, Pathologic tumor-node-metastasis; CEA, carcinoembryonic antigen; CA199; carbohydrate antigen 199; PNI, prognostic nutritional index; CONUT, controlling nutritional status.

status, which can be a favorable condition for tumor invasion and metastasis (*Liu et al., 2016*). Lymphocyte expression in tumor defense is critical by inducing cytotoxic cell death. Therefore, a decrease in the amount of such cells in the blood may be related to impaired tumor immune function, allowing for tumor progression (*Berntsson et al., 2016*; *Gooden et al., 2011*; *Jacobson, 2006*; *Tang et al., 2014*). PNI, which includes serum albumin levels and total lymphocyte count, is one of the most commonly used indicators of nutritional status (*Kanda et al., 2011*). It is known to be closely related to the prognosis of various cancers (*Mohri et al., 2013*; *Sun et al., 2015*; *Yamamoto et al., 2019*). Compared with PNI, the CONUT score includes the measurement of total serum cholesterol levels as well. Cholesterol, as an important component of cell membranes, is involved in many signaling pathways related to tumor development, progression and immunogenicity; furthermore, cholesterol levels act as an important nutritional index (*Haghikia & Landmesser, 2018*; *Jacobs et al., 2012*; *Yang et al., 2019*). Therefore, the CONUT score is considered to be a better nutritional and immune prognostic factor than PNI. In our study, PNI was found to be associated with OS and RFS of patients with PDAC after

**Table 4 Multivariate analyses offactors associated with overall survival and recurrence-free survival of PDACpatients.**

| Variable | OS | | | DFS | | |
|---|---|---|---|---|---|---|
| | HR | 95% CI | *P* value | HR | 95% CI | *P* value |
| Model 1 | | | | | | |
| Age (<52 vs. ≥52 years) | 0.98 | [0.68–1.42] | 0.933 | 0.89 | [0.61–1.30] | 0.546 |
| Tumour size (<3.1 vs. ≥3.1 cm) | 1.11 | [0.76–1.62] | 0.608 | 1.09 | [0.74–1.60] | 0.657 |
| Histopathological type (Poor vs. Medium-high) | 0.94 | [0.65–1.36] | 0.735 | 1.11 | [0.76–1.61] | 0.590 |
| Peripancreatic infiltration (Positive vs. Negative) | 0.60 | [0.38–0.94] | 0.027* | 0.60 | [0.40–0.91] | 0.017* |
| Lymph node metastasis (Positive vs. Negative) | 0.33 | [0.23–0.49] | <0.001* | 0.37 | [0.26–0.53] | <0.001* |
| Superior mesenteric artery invasion (Positive vs. Negative) | 1.07 | [0.62–1.86] | 0.800 | 1.11 | [0.64–1.93] | 0.707 |
| Invasion of portal vein system (Positive vs. Negative) | 0.85 | [0.58–1.25] | 0.411 | 0.77 | [0.52–1.14] | 0.193 |
| Nerve plexus invasion (Positive vs. Negative) | 0.85 | [0.59–1.24] | 0.408 | 0.98 | [0.68–1.42] | 0.914 |
| pTNM stage (I–II vs. III) | 1.87 | [1.30–2.70] | 0.001* | 1.63 | [1.16–2.29] | 0.005* |
| Preoperative PNI (<46.1 vs. ≥46.1) | 0.89 | [0.62–1.29] | 0.548 | 0.81 | [0.55–1.20] | 0.289 |
| Preoperative COUNT score (Low vs. High) | 4.00 | [2.82–5.67] | <0.001* | 2.93 | [2.10–4.10] | <0.001* |
| Preoperative CA199 (<36.6 vs. ≥36.6 ng/ml) | 2.23 | [1.57–3.17] | <0.001* | 1.66 | [1.20–2.29] | 0.002* |
| Model 2 | | | | | | |
| Age (<52 vs. ≥52 years) | 0.99 | [0.69–1.43] | 0.951 | 0.96 | [0.68–1.37] | 0.832 |
| Tumour size (<3.1 vs. ≥3.1 cm) | 1.11 | [0.76–1.62] | 0.593 | 1.20 | [0.84–1.71] | 0.309 |
| Histopathological type (Poor vs. Medium-high) | 0.92 | [0.64–1.33] | 0.661 | 0.94 | [0.67–1.33] | 0.723 |
| Peripancreatic infiltration (Positive vs. Negative) | 0.59 | [0.37–0.94] | 0.026* | 0.60 | [0.40–0.91] | 0.017* |
| Lymph node metastasis (Positive vs. Negative) | 0.33 | [0.22–0.49] | <0.001* | 0.37 | [0.26–0.53] | <0.001* |
| Superior mesenteric artery invasion (Positive vs. Negative) | 1.06 | [0.61–1.83] | 0.839 | 1.12 | [0.65–1.92] | 0.689 |
| Invasion of portal vein system (Positive vs. Negative) | 0.86 | [0.59–1.26] | 0.434 | 0.78 | [0.54–1.11] | 0.164 |
| Nerve plexus invasion (Positive vs. Negative) | 0.85 | [0.58–1.24] | 0.412 | 0.83 | [0.59–1.18] | 0.303 |
| pTNM stage (I–II vs. III) | 1.91 | [1.33–2.73] | <0.001* | 1.62 | [1.16–2.27] | 0.005* |
| Preoperative PNI (<46.1 vs. ≥46.1) | 0.90 | [0.62–1.30] | 0.576 | 0.94 | [0.66–1.32] | 0.701 |
| Preoperative CONUT-CA199 score (1 vs. 2 vs. 3 vs. 4) | 2.04 | [1.74–2.40] | <0.001* | 1.70 | [1.46–1.98] | <0.001* |

**Notes:**
* $P < 0.05$.
PDAC, Pancreatic ductal adenocarcinoma; HR, hazard ratio; CI, confidence interval; OS, overall survival; DFS, disease-freesurvival; pTNM, Pathologic tumor-node-metastasis; CEA, carcinoembryonic antigen; CA199, carbohydrate antigen 199; PNI, prognostic nutritional index; CONUT, controlling nutritional status.

surgery, but it was not an independent predictor; however, the preoperative CONUT score was an independent predictor of OS and RFS in patients with PDAC after surgery.

The Clavien–Dindo grading system is currently the most commonly used statistical classification system for complications. High CONUT scores correlate with an increased incidence of postoperative pneumonia, length of hospital stay and incidence of serious complications after gastric cancer (*Lin et al., 2019*). In our study, patients were divided into two groups based on the CONUT score by calculating a cutoff value and it was found that the high CONUT score group was more prone to severe postoperative complications and peripancreatic invasion, but was not more likely to develop postoperative pancreatic fistula. The reason for these results may be that the three blood indicators measured under the CONUT score, which reflect the immune and nutritional status of the body, show an increased incidence of serious postoperative complications.

However, the development of pancreatic fistula is mostly related to the hardness of the pancreas, whether is accompanied by pancreatitis, surgical technique, anastomosis type and reconstruction mode and the effects of immune and nutrition status on pancreatic fistula development seem to be insignificant.

Serum CA199 levels are a classic TMs commonly used in the management of patients with PC (*Locker et al., 2006*). This study also proved serum CA199 levels to be an independent factor that may predict postoperative survival and prognosis of patients with PDAC. However, serum CA199 levels are elevated not only in the case of PC but also in other cancers and certain inflammatory diseases. Therefore, as a diagnostic tool for PDAC, serum CA199 levels have low sensitivity and specificity (*Liu et al., 2019*; *Zeng et al., 2019*). Serum CA199 levels mainly reflects the status of the tumor, whereas the CONUT score reflects the overall status of the patient, including nutritional and immune status. We found that the combination of these two factors (CONUT-CA199 score) may provide more accurate prognostic information for patients with PDAC after surgery than either single factor, as indicated by the present ROC analyses. In addition, the CONUT-CA199 score was shown to be an independent prognostic indicator on multivariate analysis. These results suggest that the combination of serum CA199 levels and the CONUT score is more effective and provides more predictive value than serum CA199 levels or the CONUT score alone in evaluating patients with PDAC after surgery.

However, this study has some limitations. First, the sample size of our study is relatively small. According to the measure of AUCs benchmark (*Ceci & Bjork, 2000*), the AUC value is lower, this study may be related to poor specificity of CA199 regionalization related cases, inadequate sample size, source and then through joint COUNT after scoring and CA199 levels, found the AUC value is increased significantly, close to 0.7 and compared with the single use, obvious advantages, significant difference, therefore, with the enlargement of the sample size and the study population, the COUNT combined CA199 levels is expected to become effective predictor of PDAC survival in patients with postoperative prognosis.

## CONCLUSION

In summary, our study is the first to demonstrate that the preoperative CONUT-CA199 score is an independent prognostic factor for OS and RFS in patients with PDAC undergoing radical resection. As a novel, economical and reliable biomarker, the preoperative CONUT-CA199 score has potential application in the development of individualized treatments and follow-up plans.

### Funding

This work was supported by the Sichuan Science and Technology Plan Project of China (NO. 2018JY0283, 201SZYZF0015), the Luzhou Municipal People's Government-Southwest Medical University Science and Technology Strategic Cooperation Applied Basic Research Project (NO. 2018LZXNYD-ZK14), the Southwest Medical University-Luzhou Chinese

Medicine Hospital Basic Project (NO. LZZYYY2018P00039) and the Southwest Medical University Project (NO. 2018-ZRQN-077). The funders had no role in study design, data collection and analysis, decision to publish, or preparation of the manuscript.

## Grant Disclosures

The following grant information was disclosed by the authors:
Sichuan Science and Technology Plan Project of China: 2018JY0283 and 201SZYZF0015.
Luzhou Municipal People's Government-Southwest Medical University Science and Technology Strategic Cooperation Applied Basic Research Project: 2018LZXNYD-ZK14.
Southwest Medical University-Luzhou Chinese Medicine Hospital Basic Project: LZZYYY2018P00039.
Southwest Medical University Project: 2018-ZRQN-077.

## Competing Interests

The authors declare that they have no competing interests.

## Author Contributions

- Ankang Wang conceived and designed the experiments, performed the experiments, analyzed the data, prepared figures and/or tables, authored or reviewed drafts of the paper, and approved the final draft.
- Bo Sun performed the experiments, prepared figures and/or tables, authored or reviewed drafts of the paper, and approved the final draft.
- Min Wang performed the experiments, authored or reviewed drafts of the paper, and approved the final draft.
- Hao Shi performed the experiments, authored or reviewed drafts of the paper, and approved the final draft.
- Zhiwei Huang performed the experiments, authored or reviewed drafts of the paper, and approved the final draft.
- Tao He analyzed the data, authored or reviewed drafts of the paper, and approved the final draft.
- Qiu Li analyzed the data, authored or reviewed drafts of the paper, and approved the final draft.
- Jiaqi Deng analyzed the data, authored or reviewed drafts of the paper, and approved the final draft.
- Wenguang Fu conceived and designed the experiments, prepared figures and/or tables, authored or reviewed drafts of the paper, and approved the final draft.
- Yu Jiang conceived and designed the experiments, authored or reviewed drafts of the paper, and approved the final draft.

## Human Ethics

The following information was supplied relating to ethical approvals (i.e., approving body and any reference numbers):

The Ethics Committee of the Affiliated Hospital of Southwest Medical University (#KY2019053).

## Data Availability

Raw data are available in the Supplemental Files.

## Supplemental Information

Supplemental information for this article can be found online at http://dx.doi.org/10.7717/peerj.8811#supplemental-information.

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
