# Peer review of "Predictive value of CONUT score combined with serum CA199 levels in postoperative survival of patients with pancreatic ductal adenocarcinoma: a retrospective study"

_PeerJ, doi:10.7717/peerj.8811_

## Round 0.1 · original submission · Major Revisions

Both reviewers and I agree that the paper is in general the paper is well written and describes an interesting metric but the significance of the ROC curve analysis need to be fully discussed, together with the other comments from reviewer 2.

·

Basic reporting

The conceptual framework is incomplete and with some references that can be improved (15, 16, 17, 18)

Some bibliographical references are incomplete, for example, 24,35, 39.
In addition, many are old should improve this aspect.

Experimental design

The authors must describe in the methodology that they consider CONUT high and Low, as they established.
Similarly, the grouping of CA199 must be in the methodology and not in the results.

Validity of the findings

In the database in survivals there are several cases of 1 month (OS and RFS), but excluded who died within 30 days of operation. Explain this.

Additional comments

I encourage you to continue contributing to scientific knowledge, thank you for giving me the opportunity to review this manuscript.

Reviewer 2 ·

Basic reporting

In this article the authors describe the development of a novel scoring metric for pancreatic ductal adenocarcinoma that concatenates the predictive scoring power of controlling nutritional status (CONUT) score and serum carbohydrate antigen 199 (CA199) levels. In general the use of English is acceptable. Data has been shared and references are, on the whole, appropriate.But there are a couple of non-conventional uses that should be addressed:
Line 67 – Use of the word occult is not appropriate. Perhaps change to ‘unpredictable’.
Line 84 – State the CONUT score is a new scoring method. The initial paper (Ref 22) was published in 2005, almost 15 years ago. This is hardly a most recent and up to date method and the text should be addressed as such.
Table 2 - The legend needs to include full details on all statistical data displayed

Experimental design

The experimental design is well defined and outlines the research question clearly. All human particiapnt ethics are in place and there seems to be enough detail to allow replication within the text.
The statitstical methodology also seem to be approporiate. Use of the Chi-Squared test on categorical variables and ROC curve to test the predictability of the study are also suitable.
My only concern, and this needs to be justified in any response - is why was a p value of 0.05 chosen? A level of statistical significance should be chosen by the study design and not by arbitrary convention, Also, there is no mention of power within the findings. How do we know a sample size of 294 is large enough?

Validity of the findings

Leading on from the last question, levels of CA-199 have been shown to more associated with females (in an experiment with double the number of subjects):

https://bmccancer.biomedcentral.com/articles/10.1186/s12885-017-3738-y

Yet the model documented in this article does not agree. Can the authors please comment on this. Is it a facet of the difference in statistical design or a power issue?

Also, the authors need to discuss fuller the significance of the ROC curve analysis. Yes, the combined score is larger than the individual scores but at only 0.7 Swets and others have offered benchmarks for gauging AUCs, suggesting that values ≥0.9 are “excellent,” ≥0.80 “good,” ≥0.70 “fair,” and <0.70 “poor.”

Swets J A, Dawes R M, Monahan J. Psychological science can improve diagnostic decisions. Psychological Science in the Public Interest. 2000;1:1–26.

All ROC scores are pretty poor and need to be discussed further

---

## Round 0.2 · accepted · Accept

Thank you for attending to the issues raised. I have consulted with the reviewers and we are all agreed that your study is ready for publication.

·

Basic reporting

Correct

Experimental design

Correct

Validity of the findings

Correct

Additional comments

The authors have made the requested changes.

Congratulations